# Cytotoxic activity of *Ganoderma weberianum-sichuanese* isolated from the Lower Volta River Basin of Ghana against human prostate carcinoma (PC-3), leukemic T cell (Jurkat), and plasmacytoid dendritic cell (pDC)-derived acute leukemia (PMDC05) cell lines

Gideon Adotey[1]*, Raphael N. Alolga[2], Abraham Quarcoo[1], Mohammed Ahmed Gedel[1], Paul Yerenkyi[1], Phyllis Otu[1], Abraham K. Anang[3], Laud K.N. Okine[4], Winfred S.K. Gbewonyo[4], John C. Holliday[5], Vincent C. Lombardi[6]*

1 Science Laboratory Department, Accra Technical University, Accra, Ghana, 2 State Key Laboratory of Natural Medicines, Department of Pharmacognosy, China Pharmaceutical University, Nanjing, China, 3 Noguchi Memorial Institute for Medical Research (NMIMR), University of Ghana, Accra, Ghana, 4 Department of Biochemistry, Cell and Molecular Biology (BCMB), University of Ghana, Accra, Ghana, 5 Aloha Medicinals, Inc., Carson City, Nevada, United States of America, 6 Department of Microbiology and Immunology, School of Medicine, University of Nevada, Reno, United States of America

* Gadotey@atu.edu.gh (GA); vlombardi@med.unr.edu (VCL)

## Abstract

*Ganoderma* is a genus of medically important fungus that contains at least 80 species, many of which have not been properly evaluated for their anticancer potential. This study was conducted to assess the cytotoxic activity of the mycelial biomass of *Ganoderma weberianum-sichuanese* isolated from the Lower Volta River Basin of Ghana. The nuclear ribosomal internal transcribed spacer (ITS) region was analyzed to determine the phylogenetic position of this native ganoderma isolate. We then tested its cytotoxic activity against the human carcinoma cell line PC-3 (human prostate), Jurkat (human T lymphoblastoid cell line), derived from an acute T cell leukemia, and PMDC05, a plasmacytoid dendritic cell (pDC) derived from acute leukemia using the 3-(4,5-dimethyl-2-thiazolyl)-2,5-diphenyl-2H-tetrazolium bromide (MTT) assay. The ITS phylogenetic analysis demonstrated that this native Ghanaian ganoderma isolate belongs to the *Ganoderma weberianum-sichuanese* species complex. Fractions of the mycelial biomass were found to inhibit significantly (≤ 0.05%) the proliferation and survival of the three cancer cell lines, PC-3, PMDC05, and Jurkat with increasing concentrations and with $IC_{50}$ values of $27.73 \pm 5.25$, $21.31 \pm 2.40$ and $17.09 \pm 0.86$ µg/mL, respectively compared to Chang liver cells (CVCL_0238) with $IC_{50}$ value of $75.41 \pm 1.95$ µg/mL. To the best of our knowledge, these findings demonstrated for the first time, that specific constituents of *Ganoderma weberianum-sichuanese* are selectively cytotoxic to the three human cancer cell lines, suggesting

**Data availability statement:** All relevant data for this study are publicly available from the Dryad repository (https://doi.org/10.5061/dryad.b5mkkwhmq).

**Funding:** The author(s) received no specific funding for this work.

**Competing interests:** The authors have declared that no competing interests exist.

their potential efficacy in the treatment of malignancies. Future studies to isolate and characterize the biologically active molecules are warranted.

## Introduction

*Ganoderma*, an important source of fungal therapeutic agents, is widely used in ethnomedicine largely because of its immunomodulatory, antiviral, antibacterial, antioxidant, hepatoprotective, and anticancer activities [1]. Numerous health products have been developed from ganoderma mushrooms and are marketed in Asian and Western health shops [2,3]. Although the genus *Ganoderma* comprises more than 250 species worldwide, *G. lucidium* is the most studied for its chemical composition and biopharmacological properties [4]. The major bioactive compounds present in *Ganoderma* include polysaccharides, triterpenes, steroids, alkaloids, fatty acids, glycoproteins, lignin, nucleosides, nucleotides, peptides, phenols, proteins, sterols, and vitamins [5]. However, ganoderma polysaccharides and triterpenoids are the major bioactive compounds responsible for the biopharmacological activities of *Ganoderma* [5]. Indeed, several extracts from *Ganoderma* have been shown, for example, to arrest different phases of the cell cycle, leading to growth inhibition of various types of cancer cells [1].

Numerous studies have demonstrated that ganoderma polysaccharides exert anticancer activity, mainly through immunomodulation [6]. Although the mechanisms of immunomodulation are not fully known, it is believed that ganoderma polysaccharides stimulate immunological reactions through the production of various cytokines; leading to the mobilization of immune system cells [1]. In a recent clinical study, it was found that the combined use of ganoderma polysaccharides could be beneficial to cancer patients receiving conventional chemotherapy and/or radiotherapy by improving immune function as well as alleviating the toxicity of conventional therapy [6]. Although the findings in this study are very promising, the researchers recommended further clinical studies are needed to confirm the immunotherapeutic effects and related mechanisms of ganoderma polysaccharides in cancer patients.

Ganoderma triterpenoids, on the other hand, have been reported to possess hepatoprotective, antihypertensive, hypocholesterolemic, antihistaminic, antitumor, and anti-inflammatory properties [6,7]. Ganoderma triterpenoids are gaining increased recognition and are now being used as alternative adjuvants for the treatment of leukemias, carcinomas, hepatitis, and diabetes [8]. The antiproliferative effects of ganoderma triterpenoids have been demonstrated for several human cancer cell types, and the underlying mechanisms are reported to be associated with cell cycle arrest and apoptosis [8]. Thus, while ganoderma triterpenoids suppress the growth and aggressiveness of cancer cells [7], the polysaccharides stimulate anticancer responses of immune cells and also activate the production of cytokines [9]. The opposing role of ganoderma polysaccharides and ganoderma triterpenoids, with regard to the secretion of inflammatory cytokines, has made ganoderma an interesting biomedical fungus to study. Today, a comprehensive understanding of the biopharmaceutical potentials of ganoderma mushrooms from different geographical

regions is urgently needed to unlock their scientific basis for developing new drugs, and functional foods. This study was conducted to investigate the cytotoxic effect of mycelial biomass of Ganoderma LVRB-9, a novel *Ganoderma weberianum-sichuanese* species complex from the Lower Volta River Basin of Ghana on a prostate carcinoma (PC-3), a T lymphoblastic leukemia (Jurkat), and a human pDC leukemia (PMDC05) cell line.

## Materials and methods

### Fungal tissue isolation

The tissue culture of *Ganoderma weberianum-sichuanese* (Ganoderma LVRB-9) was isolated following the method described by Adotey, et al. and others outlined in [10]. Briefly, the fruiting body was surface disinfected with 70% alcohol, cut with a sterilized scalpel longitudinally, and a small piece of tissue fragment was taken aseptically from the inner core. This isolated tissue fragment was placed on an agar medium of antibiotic malt extract prepared per the instructions of the manufacturer (AMEA) (Fungi Perfecti, LLC, USA) and incubated in the dark at 28 °C for 10 days. The mycelium obtained was transferred to malt extract agar (MEA), consisting of 2% w/v malt extract, and 1.5% w/v agar without antibiotics, and cultured for an additional 10 days to obtain pure mycelium.

### Molecular identification and phylogenetic analysis

The genomic DNA of Ganoderma LVRB-9 was extracted and amplified following the methods of Aime and Phillips-Mora as outlined [11]. BLASTN, a local nucleotide database search of the ITS sequence in comparison with the reference sequences at National Centre of Biotechnology Information (NCBI) GenBank was made and the data matrices generated from the BLASTN search were subjected to ITS phylogenetic analysis by Bayesian Inference (BI) approach to establish the phylogenetic position and molecular identity [12].

### Mycelial biomass production

A solid medium of MEA was prepared in a microbox (Combinessnv, Belgium) and a layer of sterile cellophane was carefully overlaid on the agar medium. A small piece of 10-day cultured mycelium of Ganoderma LVRB-9 was placed onto the agar medium overlaid with the layer of cellophane. The microbox was hermetically sealed and incubated at 28 °C and 90% of relative humidity of air until the fungal mycelial spread through the agar medium onto the overlaid cellophane. The fungal mycelial biomass was peeled from the cellophane layer after 35 days when the primordial heads started forming out of the mycelial biomass. The mycelial biomass was dried to < 7% moisture at 50 °C with circulating air for three days and ground into a fine powder using a Mikropul model 3TH electrical mill.

### Ethanolic extraction of mycelial biomass

Three hundred grams of mycelial biomass were extracted three times with 1.5 L of absolute ethanol for six hours at 50 °C and filtered using a mixture of cotton and glass wool. The filtrates were combined and dried under vacuum with a Heidolph Rotary evaporator at 40 °C and 1 atmospheric pressure to obtain the crude ethanolic extract (**GL-C0**).

### Chromatography fractionation of ethanolic extract of mycelial biomass

The crude ethanolic extract (GL-C0) was subjected to gravity column chromatography using silica gel (60-micron particle size) as the stationary phase and varying mixtures of hexane, ethyl acetate, and methanol of increasing polarity as the mobile phase. A total of seven fractions were collected from the column. The first fraction (GL-C1) was eluted with a 100 mL hexane/ethyl acetate (90/10) mixture. The second fraction (GL-C2) was eluted with a 200 mL hexane/ethyl acetate (70/30) mixture the third fraction (GL-C3) was eluted with a 400 mL hexane/ethyl acetate (40/60) mixture. The fourth fraction (GL-C4) was eluted with a mixture of 300 mL ethyl acetate/methanol (90:10). The fifth fraction (GL-C5)

was eluted with a 200 mL ethyl acetate/methanol (50/50) mixture. The sixth fraction (GL-C6) was eluted with a 300 mL methanol/ethyl acetate (80/20) mixture and the last fraction (GL-C7) was eluted with 200 mL 100% methanol. All the fractions were concentrated under reduced pressure, using a rotary evaporator, and transferred into vials. Cytotoxic activity assay was performed on the crude ethanolic extract (GL-C0), and the seven fractions obtained. From the cytotoxic activity assay, the GL-C2 fraction displayed very promising biological activity, and therefore, was subjected to further silica column chromatography fractionation using varying mixtures of hexane, ethyl acetate, and methanol of increasing polarity as mobile phase to yield nine sub-fractions labelled as GL-C2-C1 to C9. Further cytotoxic activity analysis of these subfractions revealed that GL-C2-C1, GL-C2-C4, and GL-C2-C5 have biological activity against the cell lines used in the study. A flow chart for the silica column chromatography purification of GL-ethanol extract (GL-C0) is shown in Fig 1.

## Cell culture and cytotoxicity assay

Jurkat, PC-3, and Chang liver cells used in this study were donated by Professor Itoh of Tokyo University of Japan whereas PMDC05 was developed by Narita et al. as previously described [13]. The cells were maintained on RPMI-1640 medium supplemented with 10% fetal bovine serum, 100 U/mL penicillin, and 100 µg/mL streptomycin in 25-cm$^3$ culture flasks at 37 °C in a humidified atmosphere with 5% $CO_2$. Cytotoxic activity of extracts and fractions were tested using a 3-[4, 5-dimethylthiazole-2-yl]-2,5-diphenyltetrazolium bromide (MTT) assay. Briefly, Jurkat, PC-3, PMDC05, and Chang cells were plated at 1x10$^4$ cells/well in a 96-well plate and incubated in a $CO_2$ incubator with 5% $CO_2$ at 37 °C for 24 hours. These cells were treated with various concentrations of crude ethanolic extract (GL-C0) (1000-62.5 µg/mL), fractions from the crude extract (GL-C1 to GL-C7) (100-6.25 µg/mL), subfractions of the mycelial biomass fractions (GL-C2-C1 to GL-C2-C9) (100-6.25 µg/mL) and curcumin (36.80-2.30 µg/mL), as a positive control and the complete culture medium without extract served as a negative control. The treated cells were incubated for 72 hours at 37 °C with 5% $CO_2$ after which 50 µL of PBS solution containing 1 mg/mL MTT was added to each well in subdued light, covered with aluminium foil, and incubated for four hours. The media was discarded and 100 µL of dimethyl sulfoxide

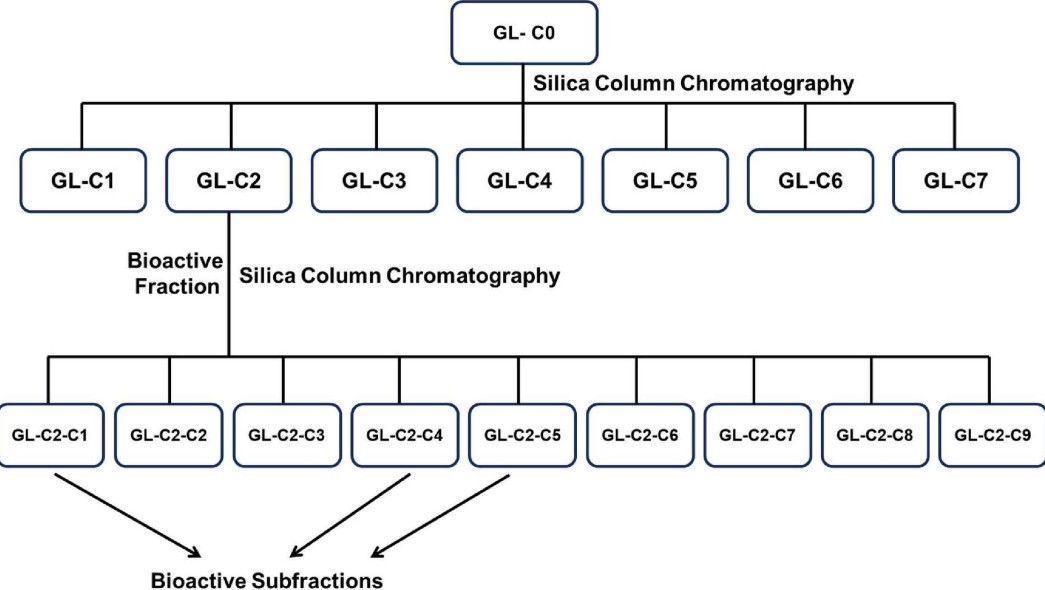

**Fig 1. A flow chart of extraction and silica column chromatography fractionation of GL-ethanol extract (GL-C0).**

(DMSO) was added to solubilize the purple formazan crystals in the dark at room temperature and the optical density was read at 570 nm with a microplate reader (Tecan Infinite M200, Austria). The experiment was performed in triplicate to determine the percentage of cell viability. The average percentage cell viability determined at each concentration was plotted as a dose-response curve and the inhibitory concentration at fifty percent ($IC_{50}$) values, a concentration inducing 50% cell inhibition, was determined from the dose-response curve by nonlinear analysis. The selectivity index (SI), a ratio of the $IC_{50}$ values of each sample in the Chang liver cell line to the $IC_{50}$ values of the other cells used in the study, was calculated. The extracts, fractions, and subfractions with SI greater than 2 were considered to have good selectivity.

### Statistical analysis

Statistical analyses were conducted using IBM SPSS Statistics version 19 (IBM Corp., USA). Given that the sample size was fewer than 30 per group, a one-way ANOVA was used, followed by Tukey's Honestly Significant Difference (HSD) post hoc test for multiple comparisons. A p-value of < 0.05 was considered statistically significant (See Supporting Information S1 Table of fractions and S2 Table of subfractions).

### Results

#### Molecular identification and phylogenetic analysis

The results of the ITS BLASTN revealed Ganoderma LVRB-9 matched with *G. weberianum* (99.24%) and *G. sichuanese* (98.73%). Bayesian analysis of the ITS sequence revealed that Ganoderma LVRB-9 clustered with *G. weberianum* and *G. sichuanese* (Fig 2) with a strong Bayesian Posterior Probability (BPP) support (99.85%), suggesting Ganoderma LVRB-9 belongs to *Ganoderma weberianum-sichuanese* species complex.

#### Cytotoxic effect of mycelial biomass crude extract and curcumin

Cytotoxic effects of the crude ethanolic extract (GL-C0) and curcumin on PC-3, Jurkat, and PMDC05 in comparison with Chang liver cell after 72 hours of treatment was investigated by MTT assay are shown in (Fig 3).

As shown in Fig 3A, the crude ethanolic extract (GL-C0) showed no observable cytotoxic effect against PC-3, Jurkat, and PMDC05 within the concentration range of 500−0 µg/mL. However, a marginal cytotoxic effect against Jurkat and PMDC05 was observed within the concentration range of 1000−500 µg/mL (Fig 3A and Table 1). The curcumin positive control, on the other hand, showed a markedly potent cytotoxic effect against all four tested cell lines in a concentration-dependent manner (Fig 3B). The most potent cytotoxic effect was against PMDC05 ($IC_{50}$ = 1.21 ± 0.09 µg/mL with selectivity Index (SI) = 6.71, followed by Jurkat ($IC_{50}$ = 2.35 ± 0.37 µg/mL, SI = 3.45) and PC-3 with $IC_{50}$ value of 4.29 ± 2.29 µg/mL, SI = 1.89 (Fig 3B and Table 1). As described in (Fig 3B and Table 1), curcumin demonstrated the lowest cytotoxic effect against Chang liver cells with an $IC_{50}$ value of 8.12 ± 0.00 µg/mL. These findings indicate that curcumin is selectively cytotoxic to PMDC05, Jurkat, and PC-3 cells but non-toxic to Chang liver cells.

#### Cytotoxic effect of mycelial biomass solvent fractions

The cytotoxic effect of the fractions (GL-C1 through GL-C7) on PC-3, Jurkat, PMDC05 and Chang liver cells following 72-hour treatment at 37 °C was investigated (Fig 4). The results revealed that fraction GL-C1 showed no cytotoxic effect on any of the four cell lines test (PC-3, PMDC05, Jurkat and Chang liver cell) (Fig 4A and Table 1). However, fraction GL-C2 (Fig 4B and Table 1) displayed a profound cytotoxic effect against Jurkat ($IC_{50}$ = 17.09 ± 0.86 µg/mL, SI = 5.85), PMDC05 ($IC_{50}$ = 21.31 ± 2.40 µg/mL, SI = 4.69) and PC-3 ($IC_{50}$ = 27.73 ± 5.25 µg/mL, SI = 3.60) when compared to Chang liver cell ($IC_{50}$ = 75.41 ± 1.95 µg/mL) (Table 1). Thus, the solvent fraction GL-C2, displayed the highest cytotoxic effect against Jurkat followed by PMDC05 and then PC-3.

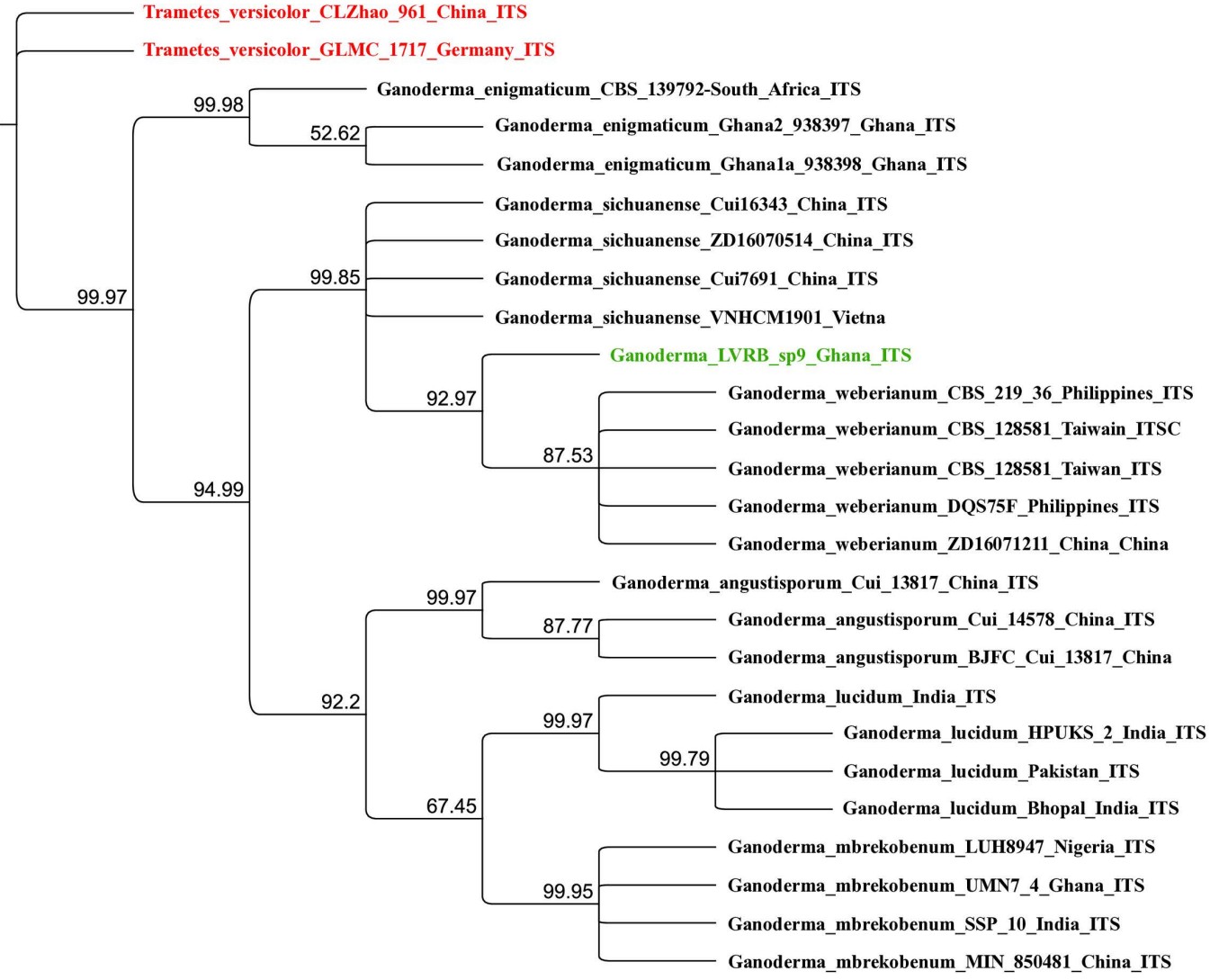

**Fig 2. Bayesian posterior probability (BPP) tree showing the phylogenetic position of Ganoderma-LVRB-9 in comparison with available ITS rDNA sequence data of Ganoderma in GenBank.** *Trametes versicolor* CLZhao 961 and *Trametes versicolor* GLMC 1717 were used as outgroups. Numbers at the branch nodes represent BPP values.

Fig 4C illustrates the cytotoxic effect of the solvent fraction GL-C3 on PC-3, Jurkat, PMDC05, and Chang liver cells. As shown in Fig 4C, GL-C3 showed no significant cytotoxic effect against PC-3 and Chang liver cells but displayed a moderately strong cytotoxic effect against PMDC05 ($IC_{50}$ = 35.69 ± 9.65 µg/mL, SI = 2.80) and Jurkat ($IC_{50}$ = 48.82 ± 5.08 µg/mL, SI = 2.05). Thus, compared to GL-C2, the cytotoxic effect of GL-C3 was generally lower ($p < 0.05$) than that of GL-C2. The solvent fraction GL-C4, GL-C5, GL-C6, and GL-C7 showed no profound cytotoxic effect against all the cell lines tested (Fig 4D–G, and Table 1).

Considering the comparatively low cytotoxic effect of GL-C2 against Chang liver cells but profound cytotoxic effect against Jurkat, PMDC05, and PC-3, it was further fractionated, and the cytotoxic effect of the corresponding subfractions tested against the four cell lines.

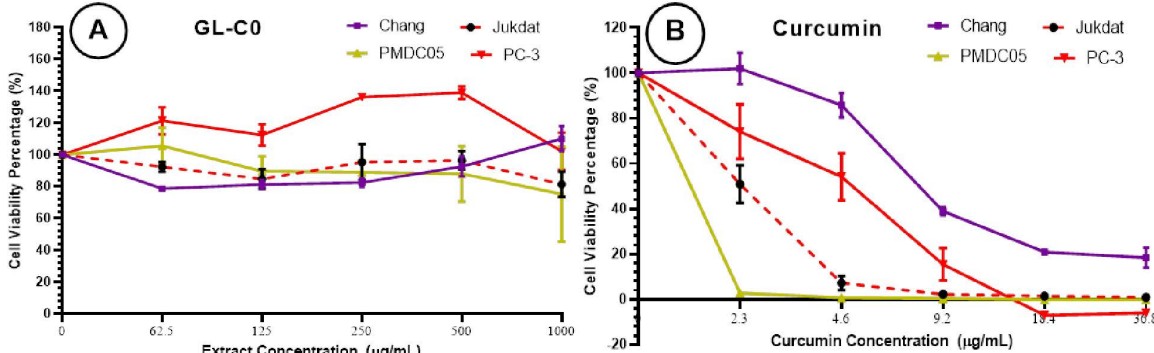

**Fig 3. Cytotoxic effect of crude (A) ethanolic extract (GL-C0) of cultured Ganoderma LVRB-9 mycelial biomass at the various concentrations (1000, 500, 250,125, 62.5 μg/mL) and (B) Curcumin as a positive control at different concentrations (36.80, 18.40, 9.20, 4.60, 2.30 μg/mL) on Jurkat, PC-3, PMDC05 and Chang liver cell lines; after 72 hours of treatment evaluated by mitochondrial activity using the MTT assay.** Each point represents the mean value of the three replicates and the error bars represent the data distribution.

**Table 1. IC$_{50}$ value and selectivity index (SI) of fractions of ethanolic extract of cultured Ganoderma LVRB-9 mycelial biomass on Jurkat, PC-3, PMDC05 and Chang liver cell line.**

| Cell line | Solvent fractions of mycelial biomass of Ganoderma LVRB-9 ethanolic extract (μg/mL) | | | | | | | | Curcumin positive control |
|---|---|---|---|---|---|---|---|---|---|
| | GC-C0 | GL-C1 | GL-C2 | GL-C3 | GL-C4 | GL-C5 | GL-C6 | GL-C7 | (μg/mL) |
| Jurkat | >1000 *0.00 | > 100 *00 | 17.09±0.86 *5.85 | 48.82±5.08 *2.05 | >100 *0.00 | >100 *0.00 | >100 *0.00 | >100 *0.00 | 2.35±0.37 * 3.45 |
| PC-3 | >1000 *0.00 | >100 *0.00 | 27.73±5.25 *3.60 | >100 *0.00 | >100 *0.00 | >100 *0.00 | >100 *0.00 | >100 *0.00 | 4.29±2.29 *1.89 |
| PMDC05 | >1000 *0.00 | >100 *0.00 | 21.31±2.40 *4.69 | 35.69±9.65 *2.80 | >100 *0.00 | >100 *0.00 | >100 *0.00 | >100 *0.00 | 1.21±0.09 *6.71 |
| Chang | >1000 | >100 | 75.41±1.95 | >100 | >100 | >100 | >100 | >100 | 8.12±0.01 |

Data are presented as IC$_{50}$ values by MTT assay from three independent experiments, performed in triplicate on Jurkat, PC-3, PMDC05 and Chang Liver cells. * Denotes selectivity index (SI).

## Cytotoxic effect of mycelial biomass solvent subfractions

The cytotoxic effect of mycelial biomass solvent subfractions (GL-C2-C1 through GL-C2-C9) was tested against PC-3, Jurkat, PMDC05 and Chang liver cells after a 72-hour treatment at various concentrations by MTT assay. The results revealed the subfraction GL-C2-C1 showed no substantial cytotoxic effect on PMDC05, Jurkat, and Chang liver cells. The subfraction GL-C2-C1, however, demonstrated cytotoxic effect against PC-3 with IC$_{50}$ value of 3.24±0.10 μg/mL and SI = 30.86 which was close to curcumin (IC$_{50}$ = 5.13±0.86 μg/mL and SI = 1.30), (p = 0.99) (Fig 5A and Table 2). The subfraction GL-C2-C2, unlike GL-C2-C1, exhibited poor cytotoxic effects against all four cell lines tested (Fig 5B and Table 2). The subfraction GL-C2-C3 (Fig 5C and Table 2), similar to GL-C2-C2, demonstrated a low cytotoxic effect against PC-3, Jurkat, and Chang liver cells but a mildly cytotoxic against PMDC05, IC$_{50}$ = 81.11±4.98 μg/mL and SI = 1.23. On the other hand, the assessment of the cytotoxic effect against GL-C2-C4 revealed the subfraction displayed markedly profound cytotoxic activity against PC-3, Jurkat, and PMDC05 but not Chang liver cells. As shown in Fig 5D and Table 2, the subfraction GL-C2-C4 exhibited the highest cytotoxic effect against PMDC05, IC$_{50}$ = 19.95±0.50 μg/mL and SI = 5.01, followed by Jurkat, IC$_{50}$ = 48.14±1.07 μg/mL and SI = 2.07 and then PC-3, IC$_{50}$ = 77.39±2.79 μg/mL and SI = 1.29. As illustrated in Fig 5E and Table 2, the subfractions GL-C2-C5, similar to GL-C2-C4, demonstrated a strong cytotoxic effect against PMDC05 with IC$_{50}$ value of 13.57±2.14 μg/mL and SI = 7.37 but a moderate cytotoxic effect against Jurkat with IC$_{50}$ value of 52.83±2.85 μg/mL and SI = 1.89.

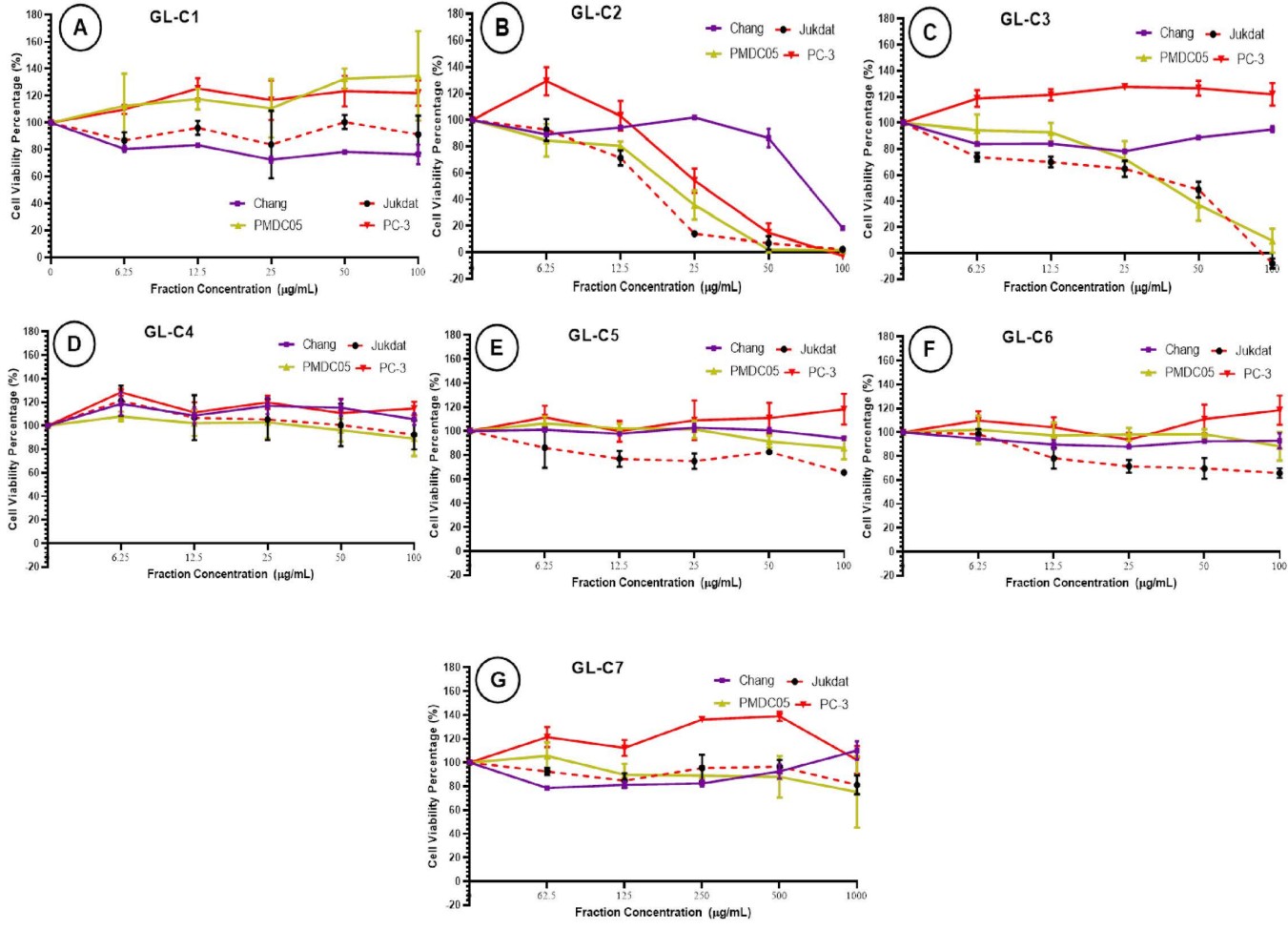

**Fig 4. The cytotoxic effect of solvent fraction (A) GL-C1, (B) GL-C2, (C) GL-C3, (D) C4, (E) GL-C5, (F) GL-C6, and (G)GL-C7) of ethanolic extract of Ganoderma LVRB-9 mycelial biomass at the various concentrations (100, 50, 25,12.5, 6.25 µg/ml) on the cancer cell lines Jurkat, PC-3, PMDC05 and Chang liver cells; after 72 hours of treatment evaluated by mitochondrial activity using the MTT assay.** Each point represents the mean value of the three replicates and the error bars represent the data distribution.

As shown in Fig 5E and Table 2, the subfraction GL-C2-C5 had no significant cytotoxic influence on PC-3 and Chang liver cells at the concentration <100 µg/mL. The subfraction GL-C2-C6 similarly displayed no cytotoxic effect against Chang liver cells (Fig 5F and Table 2) but a moderate cytotoxic effect against PMDC05 (Fig 5F and Table 2) with $IC_{50}$ value of 52.83±2.85 µg/mL and SI=1.94. The subfraction GL-C2-C7 like GL-C2-C6 showed no cytotoxic effect against Chang liver cells, Jurkat, and PC-3 but a mild cytotoxic effect against PMDC05, $IC_{50}$=88.80±4.03 µg/mL and SI=1.12. The subfraction GL-C2-C8 (Fig 5H and Table 2), similarly to GL-C2-C2, displayed no inhibitory effect on the proliferation of all the cell lines tested in this study. The last subfraction GL-C2-C9 displayed no cytotoxic effect against PC-3 and Jurkat and Chang liver cells but mildly suppressed the viability of PMDC05, $IC_{50}$=46.18±1.21 µg/mL and SI=2.16 (Fig 5I and Table 2).

## Discussions

Recently, Geng et al. (2020) reported a broad range of bioactivities for ganoderma mushrooms [14]. The reported bio-activities include antioxidation, antiinflammation, anti-liver disorders, antitumor growth, and metastasis. In the present

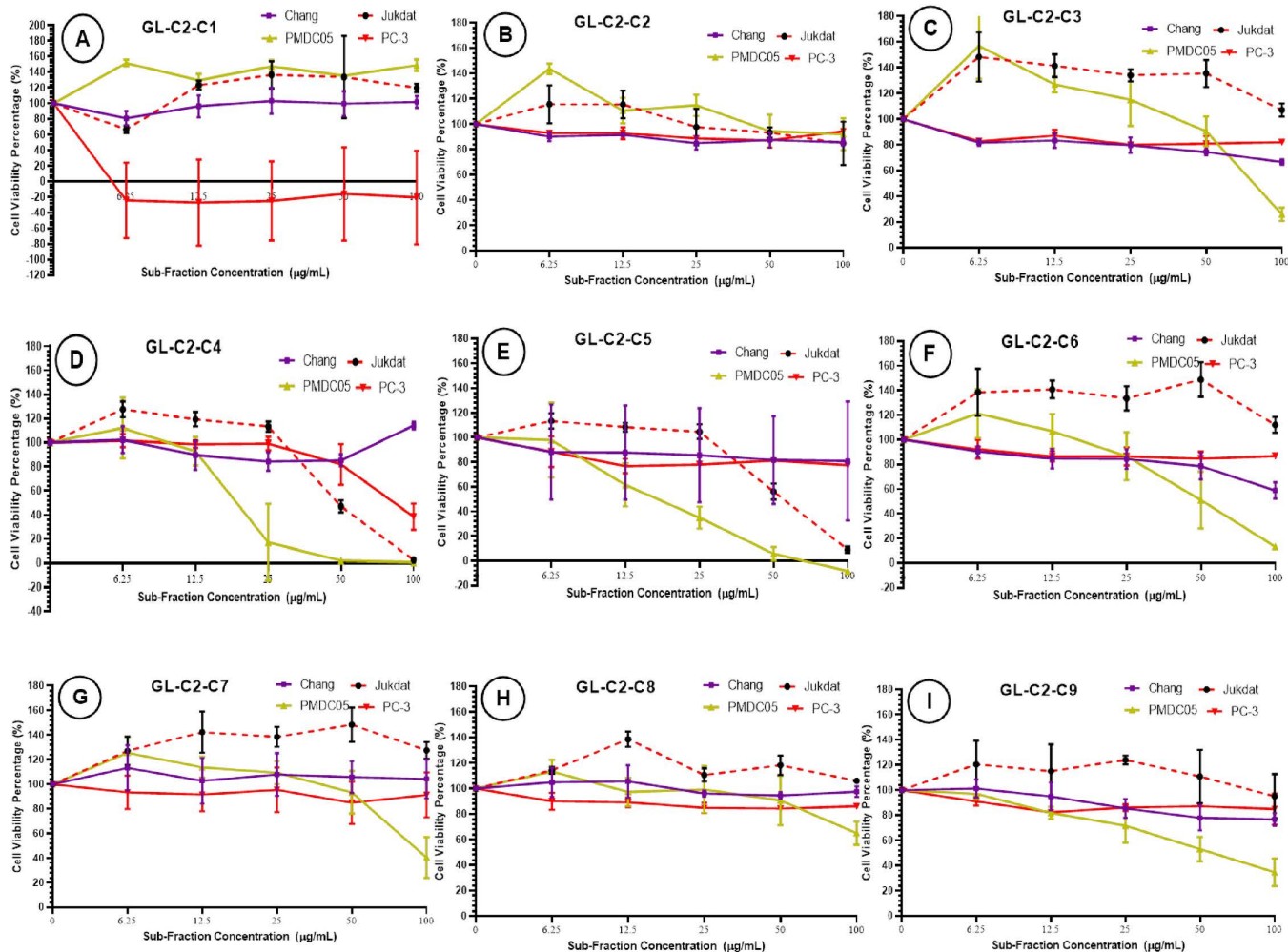

**Fig 5. The cytotoxic effect of sub-fractions (A).** GL-C2-C1, (B). GL-C2-C2, (C) GL-C2-C3, (D) GL-C2-C4, (E) GL-C2-C5, (F) GL-C2-C6, (G) GL-C2-C7, (H) GL-C2-C8 and (I) GL-C2-C9, of ethanolic extract of Ganoderma LVRB-9 mycelial biomass at the various concentrations (100, 50, 25,12.5, 6.25 µg/mL) on the cancer cell lines Jurkat, PC-3, and PMDC05, and Chang liver cells; after 72 hours of treatment evaluated by mitochondrial activity using the MTT assay. Each point represents the mean value of the three replicates and the error bars represent the data distribution.

**Table 2. IC$_{50}$ values and selectivity index (SI) of sub-fractions of ethanolic extract of cultured *Ganoderma* LVRB-9 mycelial biomass on Jurkat, PC-3, PMDC05 and Chang liver cell lines.**

| Cell line | Sub-fractions of *Ganoderma* LVRB-9 mycelial biomass ethanolic extract (µg/mL) | | | | | | | | | Curcumin control |
|---|---|---|---|---|---|---|---|---|---|---|
| | GL-C2-C1 | GL-C2-C2 | GL-C2-C3 | GL-C2-C4 | GL-C2-C5 | GL-C2-C6 | GL-C2-C7 | GL-C2-C8 | GL-C2-C9 | (µg/mL) |
| Jurkat | >100 *0.00 | >100 *0.00 | >100 *0.00 | 48.14±1.07 *2.07 | 52.83±2.85 *1.89 | >100 *0.00 | >100 *0.00 | >100 *0.00 | >100 *0.00 | 3.78±0.52 *1.76 |
| P-C-3 | 3.24±0.10 *30.86 | >100 *0.00 | >100 *0.00 | 77.39±2.79 *1.29 | >100 *0.00 | >100 *0.00 | >100 *0.00 | >100 *0.00 | >100 *0.00 | 5.13±0.86 *1.30 |
| PMDC05 | >100 *0.00 | >100 *0.00 | 81.11±4.98 *1.23 | 19.95±0.50 *5.01 | 13.57±2.14 *7.37 | 51.39±2.38 *1.94 | 88.80±4.03 *1.12 | >100 *0.00 | 46.18±1.21 *2.16 | 3.30±0.07 *2.01 |
| Chang | >100 | >100 | >100 | >100 | >100 | >100 | >100 | >100 | >100 | 6.65±0.54 |

Data are presented as IC50 values by MTT assay from three independent experiments, performed in triplicate on Jurkat, PC-3, PMDC05, and Chang Liver cells. * Denotes selectivity index (SI).

study, the cytotoxic effect of extracts and fractions of mycelial biomass of Ganoderma LVRB-9, *Ganoderma weberianum-sichuanese* species complex isolated from the Lower Volta River Basin of Ghana, on PC-3, Jurkat, and PMDC05 was investigated in comparison with Chang liver cells. Curcumin, a bioactive compound from the rhizomatous herbaceous plant *Curcuma longa*, was used as the positive control in light of its ability to inhibit the viability of many cancer cell lines, including pancreatic cancer, breast cancer, colorectal cancer, and skin cancer [15]. The results revealed that fraction **GL-C2** decreased the viability of PMDC05 with increasing concentrations and with an $IC_{50}$ value of 21.31±2.40 µg/mL compared to curcumin with $IC_{50}$ of 2.35± 0.37 µg/mL. This finding suggests that **GL-C2** may be a natural potential agent for suppressing pDCs in pDC-acute myeloid leukemia or potentially patients with chronic myelomonocytic leukemia with pDC expansion [16]. The results of the present study also demonstrated that **GL-C2** significantly (p< 0.05) affected the viability of the carcinoma cells, Jurkat, and PC-3, with $IC_{50}$ values of 17.09±0.86 and 27.73±5.25 µg/mL, respectively. The suppressive activity of **GL-C2** on the viability of Jurkat is in agreement with the work of Zhang et al. (2005) [17] and Gill et al. (2008) [18], which demonstrated that extracts from *G. lucidium* suppressed the cell viability of Jurkat in both time and concentration-dependent manner.

As a result of the suppressive activity on the viability of PC-3, PMDC05, and Jurkat, the fraction **GL-C2** was further fractionated, and the subfractions were tested. The results of the current study demonstrated that the viability of PC-3 was strongly suppressed by subfraction **GL-C2-C1** with an $IC_{50}$ value of 3.24± 0.10 µg/mL which was close to that of curcumin, $IC_{50}$ value of 5.13± 0.86 µg/mL. This finding suggests that **GL-C2-C1** may have anticancer activity against PC-3 similar to curcumin (p = 0.999) and could be useful like curcumin in developing ganoderma mushroom-based products for treating human prostate carcinoma. This current finding is consistent with the work of Jiang and others [19] which demonstrated that *G. lucidium* extract suppressed the viability of PC-3 and induced apoptosis in human prostate cancer (PC-3) cells. In another similar study, four human prostate cancer cell lines (LNCaP, 22Rv1, PC-3, and DU-145) were treated with *G. lucidum* triterpenoids (GLT) and the results showed that GLT suppresses prostate cancer cell growth by inducing growth arrest and apoptosis [20]. Since lanostane triterpenoids, such as ganoderic acid G, were detected in the mycelial biomass of ganoderma LVRB-9 in our earlier study [10], it suggests that the bioactive compounds present in **GL-C2-C1** may be similar to GLT and this may explain its growth suppressing effect of **GL-C2-C1** against PC-3 in the current study.

Although somewhat rare, blastic plasmacytoid dendritic cell neoplasms (BPDCN) are exceptionally aggressive and, when treated with conventional chemotherapies, are associated with poor outcomes [21–23]. New-generation drugs that target CD123 (the IL-3 receptor) have been developed and evaluated for treating BPDCN; however, their therapeutic efficacy is still unsatisfactory but might be improved through combined treatments [24]. It is therefore of great importance to search for biological molecules that can improve the efficacy of these drugs that target CD123. The results of the present study demonstrated that subfraction **GL-C2-C4** decreases the viability of PMDC05 with increasing concentrations and with an $IC_{50}$ value of 19.95± 0.50 µg/mL. On the other hand, subfraction **GL-C2-C5** suppressed the viability of PMDC05 with a better $IC_{50}$ value of 13.57± 2.14 µg/mL, indicating that **GL-C2-C5** has stronger suppressive activity on the viability of PMDC05 than that of subfractions **GL-C2-C4**.

Clonal expansion of mature pDCs has been reported in those with chronic myelomonocytic leukemia [25] as well as acute myeloid leukemia (AML), which is observed in approximately 5% of AML cases [16]. Depletion of pDCs may benefit these malignancies, thus, **GL-C2-C4** and **GL-C2-C5** may prove effective in this capacity; however, it should be pointed out that the PMDC05 cell line used in this study is itself derived from a pDC lymphoma so further investigations using purified primary pDCs from subjects with AML is warranted to evaluate the pDC-depleting capacity in this context.

It should also be pointed out that the MTT assay used in this study, measures cell viability as assessed by the mitochondrial ability to metabolize the substrate; however, some reports suggest that this assay may exhibit a non-specific intracellular reduction of tetrazolium which leads to underestimated results of cytotoxicity [26].

## Conclusion

This current study demonstrated that the mycelial biomass subfraction **GL-C2-C1** possesses activity against the PC-3 carcinoma cell line. Another notable finding in this study is that the subfractions **GL-C2-C4** and **GL-C2-C5** potently inhibited the viability of the PMDC05 pDC cell line. In future studies, we intend to isolate and characterize of the components responsible for the biological activity of these fractions. Indeed, the purified products of GL-C2-C1, and GL-C2-C4 and GL-C2-C5 may prove efficacious as a natural anticancer treatment for prostate and pDC malignancies, respectively, or as precursors for the development of anticancer drugs to treat these diseases.

The results of this study suggest that *Ganoderma weberianum-sichuanese* contains bioactive compounds with anti-cancer properties. Further studies will be required to characterize what compounds are responsible for these properties and if they are unique to the species under study. The identification and characterization of new compounds may prove useful in the treatment of malignancies that are difficult to treat, such as pDC lymphomas, or may serve as starting materials for the development of new drugs using rational drug design strategies.

## Supporting information

**S1 Table. Multiple Comparison IC50 and Tukey HSD of Primary Fractions.**
(PDF)

**S2 Table. Multiple Comparison IC50 and Tukey HSD of Subfractions.**
(PDF)

## Acknowledgments

We appreciate the kind support of Professor (Mrs.) Regina Appiah-Oppong, Head of Clinical Pathology Department (NMIMR). We also thank Mrs. Eunice Dotse and Mr. Ebenezer Ofori-Attah, all of the Clinical Pathology Department (NMIMR) for their invaluable assistance and encouragement. Lastly, we thank Drs. Miwako Narita and Masuhiro Takahashi for their generous gift of the PMDC05 cell line.

## Author contributions

**Conceptualization:** Gideon Adotey, Abraham K. Anang, Laud K.N. Okine, Winfred S.K. Gbewonyo, John C. Holliday, Vincent C. Lombardi.

**Formal analysis:** Gideon Adotey.

**Funding acquisition:** Gideon Adotey, Mohammed Ahmed Gedel.

**Investigation:** Gideon Adotey, Abraham Quarcoo, Mohammed Ahmed Gedel, Vincent C. Lombardi.

**Methodology:** Gideon Adotey, Raphael N. Alolga, Abraham Quarcoo, Mohammed Ahmed Gedel, Paul Yerenkyi, Phyllis Otu, Abraham K. Anang, Laud K.N. Okine, Winfred S.K. Gbewonyo, John C. Holliday.

**Resources:** Gideon Adotey, Raphael N. Alolga, Mohammed Ahmed Gedel, Vincent C. Lombardi.

**Software:** Gideon Adotey, Raphael N. Alolga, Mohammed Ahmed Gedel, Vincent C. Lombardi.

**Supervision:** Gideon Adotey, Abraham K. Anang, Laud K.N. Okine, Winfred S.K. Gbewonyo, John C. Holliday.

**Validation:** Gideon Adotey, Vincent C. Lombardi.

**Writing – original draft:** Gideon Adotey, Raphael N. Alolga, Abraham Quarcoo, Mohammed Ahmed Gedel, Paul Yerenkyi, Phyllis Otu, Vincent C. Lombardi.

**Writing – review & editing:** Gideon Adotey, Phyllis Otu, Vincent C. Lombardi.

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
