## [Decision Letter · Decision Letter 0]

PONE-D-23-32313Cytotoxic activity of Ganoderma weberianum-sichuanese isolated from Lower Volta River Basin of Ghana against human prostate carcinoma (PC-3), leukemic T cell (Jurkat) and plasmacytoid dendritic cell (pDC)-derived acute leukemia (PMDC05) cell linesPLOS ONE

Dear Dr. Lombardi,

Thank you for submitting your manuscript to PLOS ONE. After careful consideration, we feel that it has merit but does not fully meet PLOS ONE’s publication criteria as it currently stands. Therefore, we invite you to submit a revised version of the manuscript that addresses the points raised during the review process.

We look forward to receiving your revised manuscript.

Kind regards,

Mónica L. Chávez-González, PhD

Academic Editor

PLOS ONE

Journal Requirements:

Reviewers' comments:

Reviewer's Responses to Questions

**Comments to the Author**

1. Is the manuscript technically sound, and do the data support the conclusions?

Reviewer #1: Partly

Reviewer #2: Yes

2. Has the statistical analysis been performed appropriately and rigorously? 

Reviewer #1: No

Reviewer #2: Yes

3. Have the authors made all data underlying the findings in their manuscript fully available?

Reviewer #1: Yes

Reviewer #2: Yes

4. Is the manuscript presented in an intelligible fashion and written in standard English?

Reviewer #1: Yes

Reviewer #2: Yes

5. Review Comments to the Author

Reviewer #1: The article " Cytotoxic activity of Ganoderma weberianum-sichuanese isolated from Lower Volta River Basin of Ghana against human prostate carcinoma (PC-3), leukemic T cell (Jurkat) and plasmacytoid dendritic cell (pDC)-derived acute leukemia (PMDC05) cell lines" presents novel aspects on the evaluation of an extract of the fungus G. weberianum on the viability of cancer cells. However, I have some suggestions and comments to enrich the article.

1. Italicize the name Ganoderma weberianum- sichuanese in the title and in line 38.

2. Line 92, “the fruiting body was surface disinfected with 70% alcohol”.

3. Specify what type of cells are the cell lines used in the experiments.

4. How were the subfractions made? What solvent was used?

5. Was TLC performed to qualitatively differentiate the different fractions?

6. How was the calculation of the IC50 carried out, it is mentioned that through non-linear analysis, but what software was used for calculation this value?

7. In the figures it is mentioned that the mean is graphed, but are the error bars also presented?

8. It is mentioned in line 191 that there is no significant difference, but the methodology does not describe what statistical analysis was performed in each assay.

9. The effect presented is on viability (analyzed by the MTT assay), not a cytotoxic effect.

10. Improve the images, they are not properly displayed in the document due to their low quality.

11. It remains to be identified what compounds the extract contains. The introduction talks about it but its extracts lack any qualitative analysis of the presence of bioactive compounds. This is important to be able to identify the molecules responsible for the effect on the cell lines.

12. In table 2, place mL.

13. Check the mL units in the document. In some parts of the text it appears as ml.

14. Add perspectives in the conclusions, what is expected in the future with the study.

15. What was the particle size used to carry out the extraction?

16. Specify how the extract was made. What extraction technique was used?

17. I consider it more relevant to evaluate what compounds the fractions contain than to continue evaluating other cancer cells and animal models.

Reviewer #2: Some background is missing from the abstract

The authors need more to highlight the originality of the work

The authors don't explain if the extraction solvents were volatilized.

(2.3-0 36.80 μg/mL), review these data

Indicate reference to determine the selectivity index

Review the results because for the junkat cell line they report IC50 47.19 �g/mL and the graph does not show the decrease in viability for the GL-C1

Analysis is difficult due to the quality of the images

6. PLOS authors have the option to publish the peer review history of their article (what does this mean? ). If published, this will include your full peer review and any attached files.

**Do you want your identity to be public for this peer review?** For information about this choice, including consent withdrawal, please see our Privacy Policy .

Reviewer #1: No

Reviewer #2: No

---

## [Author Response · Author response to Decision Letter 1]

9 Apr 2024

Review Comments to the Author

Reviewer 1

The article " Cytotoxic activity of Ganoderma weberianum-sichuanese isolated from Lower Volta River Basin of Ghana against human prostate carcinoma (PC-3), leukemic T cell (Jurkat) and plasmacytoid dendritic cell (pDC)-derived acute leukemia (PMDC05) cell lines" presents novel aspects on the evaluation of an extract of the fungus G. weberianum on the viability of cancer cells. However, I have some suggestions and comments to enrich the article.

1. Italicize the name Ganoderma weberianum- sichuanese in the title and in line 38.

Response: Italicized accordingly throughout the document (when not used as a common noun).

2. Line 92, “the fruiting body was surface disinfected with 70% alcohol”.

Response: Thanks. Surfaced sterilized with 70% alcohol was replaced with surface disinfected with 70% alcohol as suggested

3. Specify what type of cells are the cell lines used in the experiments.

Response: The type of cells of the cell lines used are specified in Lines 33-36 and 41.

4. How were the subfractions made? What solvent was used?

Response: GL-C2 fraction was subjected to further silica column chromatography fractionation using varying mixtures of hexane, ethyl acetate, and methanol of increasing polarity to yield nine subfractions labeled GL-C2-C1 to C9. (See line 121-130)

5. Was TLC performed to qualitatively differentiate the different fractions?

Response: No, TLC was not performed in our current study because we were working with crude extracts. We primarily wanted to establish that the fractions would influence the viability of these cells in our current study. We selected the most active fractions (GL-C2-C1) because of its profound effect on Jurkat (IC50 = 17.09±0.86 μg/mL), PMDC05 (IC50 = 21.31±2.40 μg/mL), PC-3 (IC50 =27.73±5.25 μg/mL) and Chang liver cell (IC50 = 75.41±1.95 μg/mL) compared to the remaining fractions.

6. How was the calculation of the IC50 carried out, it is mentioned that through non-linear analysis, but what software was used for the calculation of this value?

Response: IC50 was calculated from the log dose-response curve using Excel.

7. In the figures it is mentioned that the mean is graphed, but are the error bars also presented?

Response: Thanks for bringing this to our attention. The correction has been made as follows ‘Each point represents the mean value of the three replicates and the error bars represent the data distribution”

8. It is mentioned in line 191 that there is no significant difference, but the methodology does not describe what statistical analysis was performed in each assay.

Response: Thanks for the comment. This is a non-quantitative observation, so we have changed this accordingly to read “no observable cytotoxic effect”.

9. The effect presented is on viability (analyzed by the MTT assay), not a cytotoxic effect.

Response: We have been following the nomenclature of the existing literature in this respect. For example, it is stated that, “Tetrazolium dye assays can also be used to measure cytotoxicity (loss of viable cells) or cytostatic activity (shift from proliferation to quiescence) of potential medicinal agents and toxic materials”. We appreciate that this may be a matter of semantics and there are limitations when using this assay so we have addressed this in the discussions. Also, please see the following references:

Mosmann T. 1983. Rapid colorimetric assay for cellular growth and survival: Application to proliferation and cytotoxicity assays. Journal of Immunological Methods. 65(1-2):55-63. https://doi.org/10.1016/0022-1759(83)90303-4

Green LM, Reade JL, Ware CF. 1984. Rapid colormetric assay for cell viability: Application to the quantitation of cytotoxic and growth inhibitory lymphokines. Journal of Immunological Methods. 70(2):257-268. https://doi.org/10.1016/0022-1759(84)90190-x

10. Improve the images, they are not properly displayed in the document due to their low quality.

Response: Image quality has been improved for all Tables and Figures as suggested.

11. It remains to be identified what compounds the extract contains. The introduction talks about it, but its extracts lack any qualitative analysis of the presence of bioactive compounds. This is important to be able to identify the molecules responsible for the effect on the cell lines.

Response: We thank the reviewer for the comment regarding the identification of compounds present in our extracts and fractions and do agree this will ultimately be important. As we continue to scientifically explore the bioactivity of these extracts and fractions, we will identify compounds responsible for the biological activities observed in these studies. However, a rigerous molecular identification of the bioactive compounds was beyond the scope of this study so we ask for the reviewer’s indulgence that this analysis will be carried out in a future study.

12. In table 2, place mL

Response: Thanks for the comment, ml was replaced with mL, the scientific notation for milliliters.

13. Check the mL units in the document. In some parts of the text it appears as ml.

Response: ml was replaced with mL throughout the document

14. Add perspectives in the conclusions, what is expected in the future with the study.

Response: Thank you, we have added additional text as suggested.

15. What was the particle size used to carry out the extraction?

Response: The particle size was 60 microns. We have added this to the methods section.

16. Specify how the extract was made. What extraction technique was used?

Response: As outlined in line 117 -120, mycelial biomass (300 g) was extracted with absolute ethanol, filtered, and the filtrates combined and dried by rotary evaporator.

17. I consider it more relevant to evaluate what compounds the fractions contain than to continue evaluating other cancer cells and animal models.

Response: Thanks for the comment. Evaluating the respective fractions on other cancer cells is a necessary step to determine if a full characterization would be justified. We agree that the purification and characterization of the bioactive compounds is important but was beyond the scope of this study and will be characterized in a follow-up study. We have articulated this is the discussions.

Reviewer #2:

1. Some background is missing from the abstract

Response: We have added additional text to the abstract to emphasize the medical importance of the Ganoderma genus.

2. The authors need more to highlight the originality of the work

Response: We have added this to the end of the abstract.

3. The authors don't explain if the extraction solvents were volatilized.

Response: Thank you, this was articulate in the methods section “dried by rotary evaporator”. We have also emphasized this with respect to each fraction.

4. (2.3-0 36.80 μg/mL), review these data

Response: Thanks. Hyphen removed.

5. Indicate reference to determine the selectivity index

Response: Thanks. As indicated in line 169, extracts, fractions or subfractions with SI greater than 2 were considered to have good selectivity

6. Review the results because for the jurkat cell line they report IC50 47.19 mg/mL and the graph does not show the decrease in viability for the GL-C1

Response: Thanks for drawing our attention to this mistake. The results have been reviewed to confirm GL-C1 did not decrease the viability of the jurkat cell line as shown in the graph.

7. Analysis is difficult due to the quality of the images

Response: The image resolution has been increased to improve quality.

---

## [Decision Letter · Decision Letter 1]

PONE-D-23-32313R1Cytotoxic activity of Ganoderma weberianum-sichuanese isolated from the Lower Volta River Basin of Ghana against human prostate carcinoma PC-3, leukemic T cell Jurkat, and plasmacytoid dendritic cell pDC-derived acute leukemia PMDC05 cell lines.PLOS ONE

Dear Dr. Lombardi,

Thank you for submitting your manuscript to PLOS ONE. After careful consideration, we feel that it has merit but does not fully meet PLOS ONE’s publication criteria as it currently stands. Therefore, we invite you to submit a revised version of the manuscript that addresses the points raised during the review process.

We look forward to receiving your revised manuscript.

Kind regards,

Miquel Vall-llosera Camps

Senior Staff Editor

PLOS ONE

Additional Editor Comments:

Please address the reviewers' remaining concerns.

Reviewers' comments:

Reviewer's Responses to Questions

**Comments to the Author**

1. If the authors have adequately addressed your comments raised in a previous round of review and you feel that this manuscript is now acceptable for publication, you may indicate that here to bypass the “Comments to the Author” section, enter your conflict of interest statement in the “Confidential to Editor” section, and submit your "Accept" recommendation.

Reviewer #1: All comments have been addressed

Reviewer #2: All comments have been addressed

2. Is the manuscript technically sound, and do the data support the conclusions?

Reviewer #1: Partly

Reviewer #2: Yes

3. Has the statistical analysis been performed appropriately and rigorously? 

Reviewer #1: I Don't Know

Reviewer #2: No

4. Have the authors made all data underlying the findings in their manuscript fully available?

Reviewer #1: No

Reviewer #2: Yes

5. Is the manuscript presented in an intelligible fashion and written in standard English?

Reviewer #1: Yes

Reviewer #2: Yes

6. Review Comments to the Author

Reviewer #1: The comments above were results, however, due to the quality of the images in the previous review, comments regarding the results were not made. On this occasion, the observations made on the results are mentioned and it is recommended to address them, since the figures lack quality for publication.

Line 105: Correct “BLASTn”.

Line 135: Correct to understand what is going to be done in the fractions. “The cytotoxic activity test was carried out on the crude ethanolic extract (GL-CO1) and the seven fractions obtained”.

Line 161: Place from highest to lowest concentration: "36.80-2.30 μg/mL".

Line 223: "Chang liver cell but a moderate cytotoxic effect on Jurkat with IC50 value 224 of 47.19±4.59 μg/mL and SI =2.12 (Figure 4A and Table 1)." This information does not correspond to what appears in Table 1.

Line 314: “This finding suggests that GL-C2-C1 may have a stronger anticancer activity against PC-3 compared to curcumin and could be useful in developing ganoderma mushroom-based products for treating human prostate carcinoma.” Was a statistical analysis performed to validate this assertion?

Line 344. This could explain the results greater than 100% viability.

Why are two IC50 values for curcumin presented in each of the tables and an average is not made for the different experiments with their 3 independent experiments carried out?

A statistical analysis remains to be performed between the fractions to determine if they present a difference between the cytotoxic effects.

It remains to be mentioned whether the data is available.

The discussion needs to be improved, if the compounds present are unknown, try to address the explanation then under the extraction conditions with the different solvents and polarities used.

In the previous review, the images could not be reviewed due to low quality, so in this review, pertinent comments are made to the figures:

Figure 1. Place GL-C0 instead of GL-ethanol extract.

Figure 3. A. Remove the comma ", PMDC05". Adjust the axes, start the X axis at 0, remove the -200 (A) and -20 (B). Check the error bars in Figure 3(B). They do not present the same format as (A).

Figure 4. The information in Figure 3 is repeated here (GL-C01). Place the Figures in the same format. Same font size in the figure title. Check that the text of the axes does not overlap. The x axis in some graphs reaches up to 150 and in others up to 120 µg/mL. If the colors of the cell type lines are the same for all graphs, a single legend can be placed for all images. Figure 4 (C) Correct pDC to PMDC05.

Figure 5. Same comment in Figure 5 regarding the format and that the text does not overlap on the axes. (A) Why is there a % viability of less than 0%? Is there any control? (H) For the PMDC05 cell line, 0 is presented at all concentrations, check if it is correct. If the colors of the cell type lines are the same for all graphs, a single legend can be placed for all images. (E) Check the error bar at the concentration of 100 g/mL for PC-3. It's correct? Since it presents a lot of dispersion in the result. (F) The error bar is not presented for PMDC05.

Table 1. Check GL-CO1, it should be GL-CO according to the text.

Reviewer #2:

Include if the same solvent ratios were used for the GL-C2 subfractions.

On line 162, include if the medium with the biomass was removed??? And if a control sample without cells was used.

Include in formula format the % to determine feasibility and SI

Include reference to compare with the Chang liver cell line if not all cancer lines are of liver origin

When the decrease in viability is less than 30%, it could be referred to as a decrease, not as toxicity, according to the ISO standard….

On line 228 specify concentration

The PC3 labels are missing in figure 5

In relation to the GL-C2-C8 sufraction in Figure 5, a viability of 0 is observed in all concentrations. Is this information correct?

On line 332 discuss the results do not rewrite the same results

Include the entire statistical part in the methodology, include a reference for the description of moderate or low in terms of the decrease in viability.

7. PLOS authors have the option to publish the peer review history of their article (what does this mean? ). If published, this will include your full peer review and any attached files.

**Do you want your identity to be public for this peer review?** For information about this choice, including consent withdrawal, please see our Privacy Policy .

Reviewer #1: No

Reviewer #2: No

---

## [Author Response · Author response to Decision Letter 2]

21 Mar 2025

Thank you for allowing us to submit this very late revision. The reviewers concerns ahve been addressed completely and point by point in the attached word document

---

## [Decision Letter · Decision Letter 2]

PONE-D-23-32313R2Cytotoxic activity of Ganoderma weberianum-sichuanese isolated from the Lower Volta River Basin of Ghana against human prostate carcinoma PC-3, leukemic T cell Jurkat, and plasmacytoid dendritic cell pDC-derived acute leukemia PMDC05 cell lines.PLOS ONE

Dear Dr. Lombardi,

Thank you for submitting your manuscript to PLOS ONE. After careful consideration, we feel that it has merit but does not fully meet PLOS ONE’s publication criteria as it currently stands. Therefore, we invite you to submit a revised version of the manuscript that addresses the points raised during the review process.

Thank you for submitting Revision 2 of your manuscript PONE-D-2332313R2. You have addressed nearly all of the reviewers’ comments, and the paper is much improved. However, before we can proceed to publication, the following points remain fundamental and must be resolved:

a)     Specify the name and version of the software used for statistical analysis in the Methods section.

b)    List all statistical tests performed and the threshold for significance (e.g., p-value).

c)     Verify and reconcile the IC₅₀ values of curcumin reported in the text against those in Table 1.

d)    Format all IC₅₀ values as, for example, “5.13 ± 0.86 μg/mL” (with spaces around “±”).

e)     Correct the spelling of the cell line “PMDC05” in the main text (it currently appears as “PMDCo5” in tables).

f)     Change all occurrences of “GL_C0” to “GL‑C0” throughout the manuscript, including Figure 1.

g)    On line 199, specify the concentration range as “500–0 μg/mL.”

h)    Clarify whether the extract alone (without cells) was included as a background control.

i)      In the Results (line 343), add the appropriate citation for the statement made in that paragraph.

Once these items have been fully addressed, your manuscript will be ready for final acceptance. Thank you for your hard work and attention to detail.

Best regards,

Karla Juarez-Moreno 

We look forward to receiving your revised manuscript.

Kind regards,

Karla Oyuky Juarez-Moreno

Academic Editor

PLOS ONE

Journal Requirements:

Additional Editor Comments:

Dear Authors,

Thank you for submitting Revision 2 of your manuscript PONE-D-2332313R2. You have addressed nearly all of the reviewers’ comments, and the paper is much improved. However, before we can proceed to publication, the following points remain fundamental and must be resolved:

a) Specify the name and version of the software used for statistical analysis in the Methods section.

b) List all statistical tests performed and the threshold for significance (e.g., p-value).

c) Verify and reconcile the IC₅₀ values of curcumin reported in the text against those in Table 1.

d) Format all IC₅₀ values as, for example, “5.13 ± 0.86 μg/mL” (with spaces around “±”).

e) Correct the spelling of the cell line “PMDC05” in the main text (it currently appears as “PMDCo5” in tables).

f) Change all occurrences of “GL_C0” to “GL C0” throughout the manuscript, including Figure 1.

g) On line 199, specify the concentration range as “500–0 μg/mL.”

h) Clarify whether the extract alone (without cells) was included as a background control.

i) In the Results (line 343), add the appropriate citation for the statement made in that paragraph.

Once these items have been fully addressed, your manuscript will be ready for final acceptance. Thank you for your hard work and attention to detail.

Best regards,

Karla Juarez-Moreno

Reviewers' comments:

Reviewer's Responses to Questions

**Comments to the Author**

1. If the authors have adequately addressed your comments raised in a previous round of review and you feel that this manuscript is now acceptable for publication, you may indicate that here to bypass the “Comments to the Author” section, enter your conflict of interest statement in the “Confidential to Editor” section, and submit your "Accept" recommendation.

Reviewer #1: All comments have been addressed

Reviewer #2: All comments have been addressed

2. Is the manuscript technically sound, and do the data support the conclusions?

Reviewer #1: Yes

Reviewer #2: Yes

3. Has the statistical analysis been performed appropriately and rigorously? 

Reviewer #1: Yes

Reviewer #2: No

4. Have the authors made all data underlying the findings in their manuscript fully available?

Reviewer #1: Yes

Reviewer #2: Yes

5. Is the manuscript presented in an intelligible fashion and written in standard English?

Reviewer #1: Yes

Reviewer #2: Yes

6. Review Comments to the Author

Reviewer #1: The authors made the relevant corrections, I only have a few observations.

1. Add to the methodology section the software used for statistical analysis, as well as the statistical tests performed?

2. Correct the spelling of the cell line PMDC05 in the text; in the tables it appears as PMDCo5

3. Correct the spelling of GL_C0 to GL-C0 throughout the manuscript.

4. Line 199, enter the range of 500-0.

5. Check the IC50 values of curcumin in the cell lines presented in the text since they do not agree with those presented in Table 1.

6. Adjust how the results are presented to IC50 = 5.13 ± 0.86 μg/mL, since in the text, the values are without spaces between them.

7. Correct in figure 1 GL-C0.

8. The paragraph on line 343 has no reference to what is described, add.

Reviewer #2: It is necessary to add the statistical analyses that were performed as well as the p parameter used.

The extract alone without cells was used as a background control?

7. PLOS authors have the option to publish the peer review history of their article (what does this mean? ). If published, this will include your full peer review and any attached files.

**Do you want your identity to be public for this peer review?** For information about this choice, including consent withdrawal, please see our Privacy Policy .

Reviewer #1: No

Reviewer #2: No

---

## [Author Response · Author response to Decision Letter 3]

15 May 2025

All the reviewers concerns have been addressed complete and without exception. The responses are attached

---

## [Decision Letter · Decision Letter 3]

Cytotoxic activity of Ganoderma weberianum-sichuanese isolated from the Lower Volta River Basin of Ghana against human prostate carcinoma PC-3, leukemic T cell Jurkat, and plasmacytoid dendritic cell pDC-derived acute leukemia PMDC05 cell lines.

PONE-D-23-32313R3

Dear Dr. Lombardi,

We’re pleased to inform you that your manuscript has been judged scientifically suitable for publication and will be formally accepted for publication once it meets all outstanding technical requirements.

Kind regards,

Huzaifa Umar

Academic Editor

PLOS ONE

---

## [Editor Report · Acceptance letter]

PONE-D-23-32313R3

PLOS ONE

Dear Dr. Lombardi,

I'm pleased to inform you that your manuscript has been deemed suitable for publication in PLOS ONE. Congratulations! Your manuscript is now being handed over to our production team.

Kind regards,

on behalf of

Dr. Huzaifa Umar

Academic Editor

PLOS ONE